# LncRNAs Regulatory Networks in Cellular Senescence

**DOI:** 10.3390/ijms20112615

**Published:** 2019-05-28

**Authors:** Pavan Kumar Puvvula

**Affiliations:** Department of Molecular and Functional Genomics, Weis Center for Research, Geisinger Clinic, Danville, PA 17822, USA; pkpuvvula@geisinger.edu; Tel.: +1-570-953-7500

**Keywords:** senescence, lncRNAs, signaling pathways, cancer, aging

## Abstract

Long noncoding RNAs (lncRNAs) are a class of transcripts longer than 200 nucleotides with no open reading frame. They play a key role in the regulation of cellular processes such as genome integrity, chromatin organization, gene expression, translation regulation, and signal transduction. Recent studies indicated that lncRNAs are not only dysregulated in different types of diseases but also function as direct effectors or mediators for many pathological symptoms. This review focuses on the current findings of the lncRNAs and their dysregulated signaling pathways in senescence. Different functional mechanisms of lncRNAs and their downstream signaling pathways are integrated to provide a bird’s-eye view of lncRNA networks in senescence. This review not only highlights the role of lncRNAs in cell fate decision but also discusses how several feedback loops are interconnected to execute persistent senescence response. Finally, the significance of lncRNAs in senescence-associated diseases and their therapeutic and diagnostic potentials are highlighted.

## 1. Background and Significance

Cellular senescence is a multifactorial progressive response exhibited by primary cells at the end of their replicative life span as was first reported by Hayflick and Moorhead [1]. The canonical view of senescence portrays it as a cellular gateway to physiological aging and age-related disorders. Aging is described as a progressive decline of tissue function and a risk factor for age-related diseases such as cancer, cardiac diseases, diabetes and neurological disorders. Senescence has been implicated in the aging process in both cellular and organismal models. Senescence and aging cells share many characteristic features, signaling pathways and molecular mechanisms. Several model studies have established an intimate link between senescence and aging/aging-associated diseases [2]. Hence, understanding the fundamental process of senescence could facilitate the effective therapeutic interventions in age-related diseases. A recent line of investigations in mouse embryonic development revamped the senescence outlook as an intrinsic tissue architectural processes in early embryogenesis and tissue remodeling process during damage and disease. Senescence response is activated by a plethora of factors including telomere attrition, genomic instability through DNA-damage, oncogenic activation, alterations of cellular secretome and dysregulated inflammatory response. Senescence is characterized by distinct metabolic and morphological alterations in nuclear-cytoplasmic compartments accompanied by characteristic neutral β-galactosidase, senescence associated heterochromatin foci (SAHFs) and senescence-associated secretory phenotype (SASP). All the senescence initiating events culminate into two interplaying and partially exclusive p53/p21 and/or pRb/p16 pathways. In the past decade, a flurry of discoveries aided the establishment of intricate transcription regulatory (TFs) networks and signal transduction pathways in senescence initiation and maintenance, of which, long non-coding RNAs (lncRNAs) were found to play various roles in a spatio-temporal and cell type specific manner and fill the gaps in signaling cascades by employing unique molecular mechanisms.

LncRNAs are endogenously encoded single-stranded RNAs with more than 200 nucleotides that lack protein coding potential. lncRNAs exhibit tissue specificity, regulated expression patterns, have a unique mode of functions and play important roles in cell proliferation, cell fate decision, apoptosis, differentiation, stem cell maintenance and division. lncRNAs mode of action is not only dependence on RNA-protein, RNA-DNA or RNA-RNA interactions but also the stoichiometry and localization of lncRNAs in a given cellular compartment. Several initial reports suggest lncRNAs aid to recruit chromatin associated complexes, histone regulators, nucleosome interactors as well as regulate epigenetics and chromatin organization, and thereby tissue specific gene expression. Furthermore, recent investigations demonstrated that lncRNAs interact with microRNAs (miRNAs) or complementary coding mRNAs and function in posttranscription regulation, alternative splicing and translocation. Based on their mechanism of action, lncRNAs are divided into signal-, decoy-, guide-, scaffold- [3] and sponge-lncRNAs [4]. A detailed review on regulatory mechanism of lncRNAs was published by Guttman and Rinn [5]. Because of their potential role in controlling gene expression of various physiological process, substantial interest was generated to explore the precise function of lncRNAs in complex human diseases. In this review, we focus on lncRNAs that have been shown to play functional roles in cellular senescence and provide a comprehensive understanding of lncRNA mediated molecular mechanisms in gene expression and their dysregulation in senescence associated disease developments.

## 2. lncRNAs in the pRb/p16 Pathway

The *INK4b-ARF-INK4a* locus encodes p15^INK4b^, p14^ARF^, and p16^INK4a^ proteins that determine cell fate and tumor growth [6]. p15^INK4b^ and p16^INK4a^ inhibit the phosphorylation of retinoblastoma protein (pRb) by cyclin dependent kinase 4/6 (CDK4/6) and drive cells to exit the cell cycle and enter into the senescence process [7]. p14^ARF^ facilitates p53 activation via MDM2 degradation and induces cell cycle arrest [8]. Hence, transcription of the *INK4b-ARF-INK4a* locus is tightly regulated [9].

Several studies have found that lncRNAs could function as scaffolds in transcription regulation of the cyclin dependent kinase inhibitor 2A (*CDKN2A*) locus. Antisense non-coding RNA in the *INK4* locus) ANRIL is a 3.8 kb lncRNA located in the antisense strand of the *INK4b-ARF-INK4a* gene cluster and shares a bidirectional promoter with p14^ARF^. ANRIL exhibits a cis mode of transcription regulation and represses the *INK4b-ARF-INK4a* locus by recruiting polycomb repressive complexes PRC1 and PRC2 and imparts heterochromatinization by layering of repressive marks of H3K27me3 and H2AK119ub1 [10]. When cells undergo senescence, transcription of p15^INK4b^, p14^ARF^, and p16^INK4a^ inversely correlates with ANRIL expression that results in loss of polycomb mediated repression on the *INK4b-ARF-INK4a* locus. A similar mode of action was recapitulated in ectopic expression of oncogenic Ras suggesting that ANRIL acts as a gatekeeper of *INK4b-ARF-INK4a* locus expression and regulates cell proliferation. Few other lncRNAs have been reported to interact with polycomb complexes and act as a scaffolding factors [11]. In proliferating fibroblasts, nuclear lncRNA-MIR31 host gene (MIR31HG) represses the gene expression of p16^INK4a^ by targeting the polycomb complex to the *INK4A* locus. Silencing of MIR31HG dislodges the repressor complex and activates p16 expression. Ectopic expression of B-Raf Proto-Oncogene, Serine/Threonine Kinase (BRAF) in human fibroblasts undergoing senescence elicits a reduction in nuclear MIR31HG levels and enforces the activation of p16^INK4a^ gene expression [12]. In breast cancer cells, elevated levels of Promoter Of CDKN1A Antisense DNA Damage Activated RNA (PANDAR) regulate the G1/S transition and partly modulate the expression of 16^INK4a^ by recruiting BMI1 [13]. Silencing of PANDAR reduces the association of BMI1 with p16^INK4a^ promoter and derepresses the transcription, suggesting that the polycomb group associates with the p16 locus in a lncRNA dependent manner.

Another lncRNA, Very long RNA Antisense to Dimethylarginine dimethylaminohydrolase 1 (VAD), exhibits a unique mode of action in *INK4* locus gene regulation. VAD has been demonstrated to regulate the *INK4* locus expression during oncogene-induced senescence by promoting removal of repressor marks H2A.Z from the promoter regions, thereby activating *INK4* locus gene expression [14].

Likewise, Myocardial infarction associated transcript (MIAT) also regulates p16 expression in breast cancer cells. Silencing of MIAT increases cellular senescence, apoptosis and decreases migration of breast cancer cells with elevated levels of p16, COX-2, miR-302 and miR-150 and downregulation of miR-29c. Gene expression analysis showed that MIAT was upregulated in estrogen receptor (ER), progesterone receptor (PR), Erb-B2 Receptor Tyrosine Kinase 2 (HER2) positive breast tumors suggesting that lncRNAs play crucial roles in cell proliferation by targeting the *CDKN2A* locus [15].

The expression of p16 can also be affected by coordinated mechanisms such as splicing, stability and translation regulation. lncRNAs have been shown to bind to mRNAs and RNA-binding proteins and influence RNA export, stability and translation. Such a case has been described for Urothelial Cancer-Associated 1 (UCA1). Regulation of p16^INK4a^ transcript stability is crucial for cell growth [16]. In proliferating cells, heterogeneous nuclear ribonucleoprotein A1 (hnRNPA1) binds to p16^INK4a^ mRNA, facilitating its rapid degradation. In senescence cells, elevated levels of lncRNA UCA1 sequester hnRNPA1 and increase the stability of p16 mRNA [17]. Exogenous expression of UCA1 in human fibroblast increases gene expressions of pro-senescence markers like transforming growth factor beta 1 (TGFβ), mitogen-activated protein kinase 14 (MAPK14), epidermal growth factor 1 (EGR1) and interleukin 6 receptor (IL6R), suggesting that UCA1 activates both autocrine and paracrine senescence signaling pathways. Recent lines of investigations suggested a positive correlation between interleukins and cytokines secreted by senescent cells and cancer development in aged cellular models and tissues [18]. As a senescence promoting agent, UCA1 has been established as an oncogene in malignant tumors such as bladder, gastric, lung, breast and hepatocellular carcinoma [19]. UCA1 expression is associated with advanced clinicopathological features, and poor prognosis for different tumor types and can be used as a potential prognostic marker and therapeutic targets [20].

An unbiased microarray study revealed that overexpression of oncogenic Ras dysregulated a large number of lncRNAs (upregulated 243 and downregulated 168) in human diploid lung fibroblasts. Among them, ANRIL and PANDA were confirmed dysregulated lncRNAs in Ras activated senescent cells by quantitative PCR [21]. Similarly, in human lung fibroblast (WI-38) undergoing replicative senescence, RNA-seq transcriptome profiling identified several antisense transcripts, pseudogene-encoded RNAs, and novel lncRNAs that were dysregulated in senescence cells [22].

These studies clearly suggest that a number of lncRNAs have been implicated in pRb tumor-suppressor pathway by directly regulating the expression of cyclins and cyclin-dependent kinases (CDKs). Cyclin D1-CDK4 complex phosphorylates retinoblastoma protein (pRb) and dissociates pRB-E2F complex, thereby releasing E2F to activate the gene expression of G1-S phase cell cycle genes. The positive feedback loop of E2F and CDKs keeps the cell cycle rolling and allows progression of G1/S and S phase, whereas in senescence, a set of lncRNAs such as VAD, MIAT1, MIR31HG, ANRIL and PANDR mediates repression of the INK4 locus and reduces the expression of cyclins. Loss of cyclins reinforces RB engagement with E2F and halts the cell cycle at G1-S transition. Hence, understanding the expression of lncRNAs and their mode of action is of paramount importance for senescence research that has broad implications in various human diseases.

## 3. lncRNAs in the p53/p21 Pathway

p53 limits the multiplication capacity of cells with DNA damage or deleterious genomic deletions and acts as a guardian of the genome. Activation of p53 is one of the stress responses and is a key initiating event in senescence. Activated p53 initiates pleiotropic events that collaboratively regulate pro-senescence gene expression. Several lncRNAs have been reported as regulators or mediators of the p53 pathway.

Long intergenic non-coding RNA-p21 (LincRNA-p21) plays key roles in senescence by targeting p21. DNA damage response (DDR) induces lincRNA-p21 through p53 activation. lincRNA-p21 acts as a scaffold to recruit hnRNP-K on p21 promoter and acts as a co-activator for p53 mediated p21 transcription regulation [23]. Mesenchymal stem cells isolated from aged mice demonstrated reduced cell proliferation, increased reactive oxygen species, and elevated levels of lincRNA-p21 compared to younger mice. Silencing of lincRNA-p21 enhanced the cell growth [24]. A similar mechanism was observed in doxorubicin induced cardiac senescence where increased expression of lincRNA-p21 interacts with β-catenin and modulates the Wnt/β-catenin signaling pathway and senescence [25].

P21 Associated NcRNA DNA Damage Activated (PANDA) is a p21-associated lncRNA directly transcribed by p53 activation. PANDA exerts its gene regulation by dynamically interact with chromatin modifiers such as PRC complex proteins and TF factors such as scaffold-attachment factor A (SAFA) and nuclear transcription factor Y subunit alpha (NFYA) and participates in senescence/cell cycle regulation. In proliferating cells, the trimeric complex PANDA-SAFA-PRC1/PRC2 acts as repressor complex and tightly controls the gene expression of pro-senescence markers like cyclin dependent kinase inhibitor 1A (CDKN1A) and interleukin-8 (IL-8). During senescence, downregulation of SAFA and PRC proteins and their interaction facilitate the dissociation of the PANDA-SAFA-PRC1/PRC2 repressor complex and activate expression of several senescence promoting genes including PANDA itself. Elevated levels of PANDA decoys NFYA and prevents its activator function in proliferating gene promoters and initiates senescence [26]. Due to its dynamic role, PANDA is found to be upregulated in gastric cancer, hepatocellular carcinoma, bladder cancer and associated with tumor size and poor prognosis [27,28]. Peng et al. reported that overexpression of PANDA promotes hepatocellular carcinoma. Mechanistically, PANDA represses inflammatory factor IL8 and suppresses senescence [29]. A recent unbiased proteomic approach in human osteosarcoma, revealed PANDA interacts with SAFA and other splicing factors like U2 small nuclear RNA auxiliary factor 2 (U2AF65), polypyrimidine tract binding protein (PTBP1) and participates in alternative splicing. PTBP1 binds to B-cell lymphoma (BCL-XS) mRNA and regulates the splicing of BCL-XS or BCL-XL variants. PANDA acts as a decoy for PTBP1 when exogenously over expressed and influences the PTBP1 splicing function. As a result, BCL-XS, which is known to be a pro-apoptotic variant, was found to be downregulated. This shift of BCL-XS to BCL-XL has been suggested to be a contributing factor for uncontrolled cell proliferation in cancer cells [30].

Maternally expressed gene 3 (MEG3) belongs to *DLK1-MEG3* imprinted locus on chromosome 14q32.3, acts as a tumor suppressor and found to be downregulated in various types of tumors and tumor cell lines. Enforced expression of MEG3 limits tumor cell proliferation. However, the induction of apoptosis or senescence depending on the cellular context. Mechanistically, MEG3 activates p53 function in three possible mechanisms. Primarily MEG3 acts as an activator and enhances p53 transcription. Secondly, MEG3 inhibits MDM2 expression and reduces p53 degradation. Finally MEG3 enhances binding of p53 to target promoters [31] and increases the p53 dependent gene expression. In human cervical cancer cells, ectopic expression of MEG3 promotes senescence through suppression of MDM2 expression and subsequent activation of p53 [32,33].

P53 Induced Noncoding Transcript (Pint) is another p53 responsive lncRNA targeting PRC2 complex that negatively regulates TGF-β, MAPK and p53 autoregulatory pathways [34]. Likewise, P53 Regulation Associated LncRNA (PRAL) regulates p53 and inhibits cell proliferation. When overexpressed in lung cancer cell lines, PRAL inhibited cell proliferation by activating p53 transcription [35]. Hence, absence of PRAL expression or deletion of 17p13.1 locus, which encodes PRAL, was often found in human cancers correlated with PRAL tumor suppressor function.

Cytoplasmic lncRNAs interact with RNA-binding proteins and participate in mRNA translation regulation. Such an example comes from a 7SL lncRNA study. p53 translation is under the tight control of a competitive function between human antigen R (HuR) and 7SL lncRNA. These two factors directly compete for binding to the 3′ UTR region of p53 mRNA. In proliferating cells, elevated levels of 7SL displace HuR from 3′UTR region of p53 and reduces its translation efficiency. In contrast, downregulation of 7SL promotes HuR binding to p53 3′UTR, translation and protein production [36]. Similarly, p53 translation is under the tight control of LINC00673, a highly abundant lncRNA in lung adenocarcinoma tissues. Depletion of LINC00673 induces G1- to S-phase arrest and triggers senescence response in lung cancer cells. Inhibition of p53 rescues the LINC00673 mediated senescence. Functional studies established that LINC00673 negatively regulates the p53 translation and affects the proliferation of lung cancer cells [37].

In 2394 tumor specimens from 12 cancers, a genome-wide survey identified frequent somatic mutations and copy number alterations in widely expressed lncRNAs. A genetic screen identified Focal Amplified LncRNA On Chromosome 1 (FAL1) as an oncogene, and functional studies proved that FAL1 exerts its oncogenic activity via negative regulation of p21 expression. Silencing of p21 transcription by FAL1 depends on its direct interaction with a polycomb1 complex protein, BMI1. FAL1 increases BMI1 stability and regulates the association of BMI1 with promoter regions of large number of genes including CDKN1A. Loss-of-function of FAL1 induced senescence in fibroblasts and regressed the ovarian tumor growth in vivo [38].

Shi et al., reported that silencing of BRAF-Activated Noncoding RNA (BANCR) contributes to the translation regulation of p21 protein. When overexpressed, BANCR significantly reduced the growth of colorectal cancers through negative regulation of p21 protein synthesis [39,40].

Ovarian Adenocarcinoma Amplified lncRNA (OVAAL) is an up-regulated lncNRA in ovarian, endometrial, colorectal cancers and melanomas. Depletion of OVAAL reduces tumor growth and inhibits cancer cell proliferation. Mass spectrometry results revealed that OVALL preferentially binds PTBP, an RNA-bonding protein known to promote the translation of p27 and p21 by directly binding to 5′UTR regions of p27 mRNA [41]. Knockdown of OVALL induces senescence by enhancing the gene expression of p21 and p27. Mechanistically, OVALL competes with p27 mRNA for binding to PTBP1. Hence, loss of function of OVALL resulted in PTBP association with p27mRNA. In contrast, exogenous expression of OVALL reversed the association between PTBP1 and p27 mRNA and decreased the cellular levels of p27 protein [42].

In mouse fibroblasts undergoing replicative senescence (RS), a microarray study identified 289 dysregulated RNAs in senescent cells. Among them, AK156230 was found to be reduced significantly. When downregulated, lncRNA-AK156230 loss of function promoted cells to undergo senescence response by activation of p53, p21 and downregulation of cyclin dependent kinase 1 (CDK1). Gene expression studies revealed that authophagy promoting genes such as Unc-51 Like Autophagy Activating Kinase 2 (ULK2), Autophagy related 7 (ATG7), Autophagy Related 16 Like 2 (ATG16L2) and cell cycle genes such as CDKN1A, Reprimo, TP53 Dependent G2 Arrest Mediator Homolog (RPRM), Cell Growth Regulator With EF-Hand Domain 1 (CGREF1) contributed to the activation of senescence [43].

Here, we conclude that several lncRNAs exhibited the functional correlation with p53 in senescence by directly or indirectly modulating the p53 mediated signaling pathways at the level of transcription or translation. Of significant interest, recent publications highlighted the role of individual lncRNAs in p53 regulatory networks and how the cross talk between lncRNAs and p53 function was perturbed in cancer [44]. It is likely that the large repertoire of lncRNAs might be contributing to the function of p53 as a guardian of the genome. Hence, further characterization of p53-lncRNAs relation in senescence will facilitate not only the molecular understanding of lncRNA mediated gene regulation but will also provide a broader understanding of lncRNAs in DNA damage response and genome stability, which have direct implications in cancer and age-related diseases.

## 4. lncRNAs in Telomere Regulation

Replicative senescence (RS) is a manifestation of telomere erosion. Loss of telomerase [45] and shortening of telomeric repeats are major causes of telomere erosion [46]. Telomere shortening is one of the characteristic features of senescent cells. Telomeres encode a specific telomere repeat-containing lncRNAs called telomerase RNA component (TERC) and telomeric repeat-containing RNA (TERRA). Telomere structure is maintained by shelterin components telomeric repeat factor 1 (TRF1) and telomeric repeat-binding factor 2 (TRF2). TERRA mediates recruitment of heterochromatin protein 1 (HP1α), hnRNPA1, shelterin components and protects chromosome ends [47]. TERC contributes to maintenance of telomere length and heterochromatin assembly. Overexpression of TERRA bypasses premature senescence and aging in murine models [48], whereas abnormal expression of TERRA induces premature senescence in fibroblasts due to suppression of telomere elongation [49]. TERRA is encoded by chromosome 20Q region and aids polycomb recruitment and layering of H3K9me3, H4K20me3 and H3K27me3 heterochromatin marks at telomers [50]. Similarly, GUARDIN, a p53 responsive lncRNA, plays a pleotropic role in DNA repair and telomere maintenance. Silencing of GUARDIN removes pro-survival mechanism and leaves anti-survival signaling intact. As a result, apoptosis was found to be induced and senescence was initiated in GUARDIN loss of function cells. Mechanistic understanding reveled that GUARDIN protects the telomere ends from DDR through TRF2 activation by negatively inhibiting miR-23a activity. TRF2 and GUARDIN loss-of-function recapitulates ectopic expression of miR-23 on telomere dysfunction, suggesting that GUARDIN acts as an endogenous competitive lncRNA for miR-23a and positively regulates TRF2 expression. GUARDIN functions through BRCA1 mediated homologous recombination (HR) and non-homologous end joining (NHEJ) mechanisms. Primarily, GUARDIN activates the transcription of breast cancer type 1 susceptibility protein (BRCA1) expression. Secondly, GUARDIN enhances the heterodimerization of BRAC1 with BRAC1 associated RING domain 1 (BARD1), which plays important roles in repairing double-strand breaks. Due to the cooperative function of TRF2 and BRCA1, GUARDIN was reported to maintain the de novo structure of DNA and genome integrity [51].

## 5. lncRNAs in Stress Responses

Several extrinsic factors have been reported to affect the cellular function and eventually senescence phenotype in culture conditions [52,53,54,55,56]. Few studies have found a direct link between lncRNAs and UV-radiation induced photodamage senescence response. In fibroblasts treated with UVB irradiation, metastasis associated lung adenocarcinoma transcript 1 (MALAT1) was identified as significantly upregulated lncRNA accompanied by matrix metalloproteinase-1 (MMP-1) secretion and senescence phenotype. Silencing of MALAT1 in UVB irradiation rescued the senescence phenotype by dampening MMP-1 secretion and downregulating the ERK/MAPK pathway, but not in the UVB-induced reactive oxygen species (ROS) pathway [57]. lncRNA CDID-2:1 participates in UVB induced senescence through generation of ROS in melanocytes [58]. Interestingly, in UVB-irradiated primary human dermal fibroblasts, RNA-seq profiling identified RP11-670E13.6 as an upregulated lncRNA. When downregulated, RP11-670E13.6 induced robust senescence response in UVB-irradiated cells through activation of the p16-pRb pathway, suggesting that lncRNA acts a protective factor in delaying senescence response in UVB damaged fibroblasts [59]. H_2_O_2_ induces senescence through ROS induction and has been reported to be one of the oxidative stressors. A recent study combining transcriptome profiling and loss-of-function screen in human fibroblasts identified lncRNA-OIS1 as differentially upregulated RNA. lncRNA-OSI1 gene expression is correlated with dipeptidyl peptidase 4 (DPP4) expression. Inhibition of lncRNA-OIS1 reduces the expression of CDKN1A and bypasses senescence response through downregulation of DPP4. However, the exact mechanism of how lncRNA-OSI1 controls DPP4 expression was not identified [60].

## 6. lncRNAs in Feedback Regulation

Positive and negative feedback loops counteract each other and influence the outcome of senescence response. Cytokine-induced feedback loops are classical examples of positive feedback mechanisms reported for SASP-mediated paracrine regulation of senescence [61]. On the other hand, miRNAs have negative feedback mechanisms to control the persistent inflammation mediated by nuclear factor kappa-light-chain-enhancer of activated B cells (NF-kB) activation [62]. Recent studies demonstrate that lncRNAs activate both positive- and negative feedback loops and participate in prolonged senescence response. HOTAIR primarily functions as a scaffolding factor for the recruitment of the PRC1/2 and LSD-CoREST complex to repress targets genes in nuclear compartment [63]. Several independent lines of evidence solidified the oncogenic role of HOTAIR in solid tumors [64]. Enforced expression of HOTAIR accelerate cell proliferation, growth, migration and metastasis in tumor cells and increases chemo resistance. Overexpression of HOTAIR in ovarian cancer cells activates the senescence phenotype through DNA damage response, activation of NF-kB/IL-6 and Chk1/p53/p21 pathways and contributes to the chemoresistance [65]. Interestingly, DDR induced NF-kB binds to the HOTAIR promoter and activates its transcription. This positive feedback loop enforces sustained NF-kB activation. HOTAIR-induced IL-6 secretion initiates a second cascade of anti-apoptotic events by overexpression of B-cell lymphoma 2 (BCL-2) and BCL-XL. Hence, sustained IL-6 secretion contributes to chemoresistance and senescence. These studies support the emerging concept of senescence as tumor promoting process through activation of pro-inflammatory responses in tumor microenvironments [64]. HOTAIR also functions as a scaffold for ubiquitination. During proliferation, the HuR protein binds to HOTAIR and reduces its levels presumably due to post-transcriptional decay of HOTAIR mRNA. The mechanism underlying the HOTAIR instability likely involves targeting of the 1142–1272 nucleotide region of HOTAIR by HuR and let7/Argonaute2 (Ago2) complex and decreases the half-life and steady state transcript levels. In human fibroblasts undergoing senescence, elevated levels of HOTAIR and reduced levels of HuR force HOTAIR to differentially interact with the RNA component of E3 ubiquitin ligase [66]. However, the contribution of ataxin-1 and snurportin-1 towards senescence remains unknown. As summarized in Figure 1, a growing number of lncRNAs participate as effectors/mediators/ regulators of the p53/p21 and/or pRb/P16 pathway and are implicated in senescence promotion or inhibition. Akin to HOTAIR, ANRIL participates in the NF-kB/ANRIL/YY1+ANRIL/IL-6/NF-kB positive feed-back loop for constitutive activation and maintenance of SASP mediated senescence response in auto- and paracrine mechanisms. Likewise, lincRNA-p21 participates in the p53/lincRNA-p21/hnRNPK+p53+lincRNA-p21/p21feed-forward loop to activate p21 expression and maintenance of the senescence phenotype. In contrast, p53/lnc-ROR: p53/Pint participate in negative feedback loops to maintain cellular homeostasis. These functional intricacies between lncRNAs and their interacting/regulating cellular factors determine the feedback regulation of senescence gene expression. 

## 7. Other lncRNAs

Compiling evidence has established the significance of lncRNAs in various functional process. Examples highlighted in this section illustrate the broad spectrum of mechanistic functions of lncRNAs ranging from translation regulation to protein modifications in senescence and how they control cell fate decision.

MALAT1 is upregulated in several cancer types and acts as a pro-proliferative gene. Recent studies showed that reduction of MALAT1 impairs cell proliferation. Initial evidence suggests that MALAT1 was involved in E2F transcription factor 1 (E2F1) mediated transcription regulation of cell cycle genes in fibroblast (HDFs). However, Tripathi et al., conducted a time-course experiment in MALAT1 depleted WI38 cells and showed that MALAT1 depletion immediately activates the p53 pathway that precedes pRb dephosphorylation, E2F inactivation and reduced expression of E2F target genes. These observations implicate a crucial role of p53 in MALAT1-mediated cell proliferation. In addition, silencing of MALAT1 not only compromised G1-S transition but also perturbed mitotic progression. Functional studies revealed that reduced levels of cellular Myb-related protein B (B-MYB), perturbation of B-MYB alternative splicing and dysregulation of B-MYB transcriptional activity were the main effectors leading to aberrant mitotic defects in MALAT1 depleted fibroblasts [67].

**SIRT1 Antisense RNA**: SIRT1 is a NAD^+^-dependent protein deacetylase and plays an important role in longevity and cellular senescence. Overexpression of SIRT1 could rescue the oncogene induced senescence (OIS) [68]. In endothelial progenitor cells (EPS), Nicotinamide Phosphoribosyltransferase (NAMPT) treatment enhances the expression of SIRT1 and SIRT1AS lncRNA. Overexpression of SIRT1AS directly contributes to the elevated levels of SIRT1. Bioinformatic and luciferase analysis suggest that SIRT1AS negatively regulates miR-22 activity and relives miR-22 mediated repression on SIRT1. These results demonstrate that overexpression of SIRT1AS increases proliferation and migration of EPS through functions as SIRT1AS/miR-22/SIRT1 Axis [69].

**As-UCHL** is another lncRNA, which mediates p53 pathways through regulation of ubiquitin C-terminal hydrolase L1 (UCHL1) mRNA translation [70]. Ectopic expression of UCHL induced senescence with increased p14ARF, p53, p27kip1 and decreased MDM2 expression [71].

**H19:** Chromosome 11p15 locus encodes a paternally imprinted and highly conserved lncRNA RNA called H19 and plays important roles in tumorigenesis. p53 and H19 exhibit counter-regulating mechanisms as evidenced by the repressor activity of p53 on H19 promoter [72]. Conversely, H19 directly interacts with p53 and inhibits p53 function in gastric cancer [73]. In hypoxic stress, H19 was found to be constantly upregulated by hypoxia inducible factor 1 subunit alpha (HIF1a) transcription factor. When overexpressed under hypoxic conditions, H19 promotes cell proliferation that is in part mediated by direct inhibition of CDKN1C and activation of cyclin E2 [74] gene expression. H19 promotes tumor cell survival under adverse conditions. In nucleus pulposus cells (NPCs) undergoing senescence through H_2_O_2_ treatment, H19 was significantly upregulated. Overexpression of H19 accelerated H_2_O_2_ induced the senescence phenotype through increased levels of A disintegrin and metalloproteinase with thrombospondin motif 5 (ADAMTS-5), MMPs and collagen 1 content. Functionally, H19 binds to miR-22 and derepresses the target gene, β-catenin, and activates the Wnt/β-catenin signaling pathway and enforces a senescence cascade [75]. Xu et al., reported that activation of β-catenin induces DDR accompanied by upregulation of γ-H2A.X mark, p16, p53 and p21 in mesenchymal stem cells [76,77]. An interesting study on the role of H19 on aged endothelial cells in mouse models demonstrated that H19 expression was negatively correlated with the proliferative potential of endothelial cells. Loss of H19 facilitates senescence and pro-inflammatory response by activating the IL-6/STAT3 pathway. Mechanistically, depletion of H19 increases IL-6 expression, phosphorylation of signal transducer and activator of transcription 3 (STAT3) and its downstream targets like p21 and vascular cell adhesion protein 1 (VCAM-1) expression. These reports implicate H19 as a key mediator with both activator and repressor functions and influences the cell cycle [78].

**ASncmtRNA-2:** In endothelial cells undergoing RS, RT-PCR analysis identified ASncmtRNA-2 as a significantly upregulated lncRNA companioned by elevated levels of p16. Similar expression pattern was observed in aortas of old mice. When overexpressed in proliferating endothelial cells, ASncmtRNA-2 induced a stable growth arrest at G2/M phase with accumulation of miR-4485 and miR-1973 and the replicative senescence phenotype [79]. In breast cancer cells, miR-4485 has been demonstrated to modulate the mitochondrial complex 1 activity, ROS levels and caspase-3/7 activation, and tumorigenicity in nude mouse model [80] suggests that ASncmtRNA-2 might act via miR-4485/ROS axis in RS response.

**Senescence-associated long non-coding RNA (SALNR):** In human fibroblasts undergoing RS, a microarray screening identified SALNR as significantly downregulated in senescent cells. When overexpressed, SALNR delayed the onset of cellular senescence accompanied by decreased expression of both p16 and p53 proteins. Mechanistically, SALNR regulates senescence by interacting with RNA-binding protein NF90 and prevents the senescence-associated miRNA biogenesis. During oncogenic stress, translocation of NF90 to the nucleolus ameliorates its ability to inhibit senescence-associated miRNA biogenesis [81].

**TERRA:** Poly (ADP-ribose) polymerase-1 (PARP1) is a sensor of DNA damage and regulates OIS through stimulation of SASP. Functionally, PARP mediated SASP regulation is associated with the interaction between PARP and lncRNA-telomeric repeat-containing RNA (TERRA). Moreover, melatonin treatment attenuated the SASP production by altering the interaction between PARP and TERRA [82], suggesting that TERRA might act as a mediator in PARP regulated gene expression.

**VIM-AS1** is an antisense lncRNA found to be upregulated in colorectal cancer (CRC). Overexpression of VIM-AS1 upregulated lymph node metastasis and vascular invasion. Conversely, silencing of VIM-AS1 led to cellular senescence and could be used as novel target for CRC treatment [83].

A recent study on PBMCs from type 2 diabetes reported significantly increased levels of HOTAIR, MEG3, LET, MALT1, MIAT, ANRIL, XIST, PANDA, GAS5, Linc-p21, ENST00000550337, PLUTO and NBR2 along with established senescence markers such as p53, p21 and β-galactosidase, suggesting the clinical relevance of lncRNAs and senescence in human diseases [84].

As summarized in Figure 1, lncRNAs like H19, MALAT1, UCA1, lincRNA-p21 and PANDA function on multiple targets in different cellular compartments in order to maximize their functional consequence on downstream pathways of senescence. These studies not only invigorate our understanding of senescence but also prompt us to investigate and design new strategies to combat complex disorders.

## 8. Conclusions and Future Directions

There is clear evidence that lncRNAs playing important roles in various biological processes and add another layer of complexity to gene regulation. Under the current understanding, several of the missing links in or among the existing molecular pathways have been interconnected with lncRNAs. Hence, perturbation, or dysregulation of lncRNAs is reported to cause cellular imbalance and result in disease manifestation. However, the precise role and impact of lncRNAs in disease initiation or maintenance remain obscure. Most of the experimental approaches on lncRNAs have been concentrated on genome wide profiling studies between healthy and disease patients. These profiling studies uncovered thousands of dysregulated lncRNAs, and co-expression network analysis provides evidence that lncRNAs participate in disease associated signaling pathways [85,86]. Unlike protein coding genes, poor conservation, low level of expression, lack of functional studies and limited computational algorithms restricted our understanding of novel lncRNAs. However, few algorithms such as HLPI-Ensemble [87], LPI-NRLMF [88] and IRWNRLPI [89] have been developed to predict lncRNAs-protein interactions in humans [90]. Hence, better tools are needed to explore the structure–function relationship of lncRNA and their therapeutic potential in various diseases. LncRNA databases such as LNCipedia (http://www.lncipedia.org) [91], lncRNAdb (http://www.lncrnadb.org/) [92], LncRNAwiki [93], NONCODE (http://www.noncode.org) [94], LncRNABase [95] and TANRIC [96] have been established to identify novel lncRNAs, to provide comprehensive annotation, to study the expression, to discover the function through target prediction. Due to significant associations between lncRNAs and human diseases, KATZLDA [97] developed an algorithm to predict lncRNAs association with diseases and aids to construct functional similarity gene networks [98,99,100]. Disease-specific lncRNA expression has been updated in the lncRNA disease database (http://cmbi.bjmu.edu.cn/lncrnadisease). By using these tools, in the past two years, several reports have been published based on the expression analysis of lncRNAs and their nearby genes. Functional pathways of enrichment analysis of nearby genes of lncRNAs provided the signaling mechanisms coregulated by lncRNA expression and also highlighted the cis-acting functions of a given set of dysregulated lncRNAs. These kinds of co-expression analysis failed to address how lncRNAs regulate the expression of distinctly located target genes and their trans-acting molecular mechanisms. Hence, there is necessity to introspect the limitation and scope of in silico analysis and new tools should be implemented to decipher the trans-acting functions of lncRNAs. Secondly, bioinformatic predictions should be complemented with functional validations in a relevant cellular model. Discovering a novel function of a lncRNA should be carried out in an unbiased approach by integrating 3 factors (1) lncRNA dependent gene expression, (2) lncRNA bound targets information and (3) lncRNA mediated cellular or morphological changes in a relevant cell type of tissue by employing loss- or gain of function experiments.

Recent studies on lncRNAs raised some controversies in molecular functions of well-known lncRNAs such as MALAT1, LincRNA-P21 and EVF2 [101,102]. Their opposing mechanism of actions, despite the same lncRNA loss of function, attracted intense scrutiny from researchers around the world. Comparative analysis of the experimental procedures employed in deactivating or inactivating the same lncRNA cleared the air, suggesting that the opposing phenotypic differences are due to off-target effects per se in RNAi based inactivating strategies. Genetic rescue experiments unambiguously established a novel tumor suppressor role for MALAT1 by toppling the decade-old claim of MALAT1 being an oncogene [103] in breast cancer. These studies not only highlight the negative impact of off-target effects in RNAi knockdown strategies but underscore the implementation of meticulous experimental procedures and alternative supporting evidences. With similar opinion, recently, three reviews further highlighted the caveats of lncRNAs research and conclusively demonstrated that the majority of the anticancer lncRNA studies were invalidated due to off-target effects in loss of function studies and having insufficient genetic rescue experiments. Hence, rigorous investigations are needed to address the tissue specific roles of lncRNAs.

Because of the tissue specific expression, dynamic alterations upon disease in body fluids such as blood, plasma, urine and their aberrant expression in circulating tumor cells, lncRNAs could be considered as next generation biomarkers or targets for human diseases. Several reviews highlighted that expression of lncRNA is a direct indicator of the severity of the disease. These unique properties render lncRNAs as potential noninvasive biomarkers and therapeutic targets for human diseases.

Though some lncRNAs express in low copy number, their unique expression pattern in a given subtype of cancer provides an added advantage for the detection of subtype-specific lncRNA-based biomarker. The presence of many structural regulatory sites in lncRNAs provides opportunities to develop novel structure-based drugs. Different strategies have been enumerated to modulate lncRNA expression. To name a few, use of synthetic molecules/peptides to block the lncRNA interaction, aptamers against specific structural regions of lncRNAs and GapmeRs antisense oligos are effectively tested for silencing of lncRNAs in vivo. In contrast, viral-mediated gene delivery, dextran nanoparticles and RNA mimics were employed for therapeutic overexpression of lncRNAs.

In spite of several obstacles, lncRNAs research has reached pre-clinical and clinical stages in cancer models. LncRNAs such as HOTAIR, MALAT1, GAS5, PC3, and H19 are aberrantly regulated in various malignant tumors and associated with carcinogenesis, metastasis, and prognosis. The Food and Drug Administration (FDA) approved PCA3 as more specific and urine biomarker lncRNA for prostate cancer. Similarly, H19 targeted therapy is under phase I/II clinical trials in bladder, pancreatic and ovarian cancer patients.

In addition, circulating HOTAIR was demonstrated to be useful for diagnosis of breast cancer, bladder and colorectal cancer. Similarly, MALAT1, UCA1, ANRIL, and NEAT1 were shown to predict lung cancers. LncRNAs H19 could be used as a valuable biomarker for HCC and bladder cancer. Apart from the biomarker roles, few studies exploited the role of lncRNAs in improving therapeutic sensitivity. Knockdown of HOTAIR enhances the sensitivity of cancer cells to chemotherapeutic agents like cisplatin and doxorubicin. Similarly, knockdown of TUG1 enhanced the chemo sensitivity of lung cancer cells. GAS5 and MALAT1 modulate chemo resistance in gastric cancers and glioblastoma multiforma cells, respectively. LncRNAs demonstrated promising results in diagnostic, prognostic and therapeutic trials against malignant tumors, but efficient delivery strategies and validations are required for their clinical implication.

In autoimmune diseases, lncRNA research has been at the infancy stage. Given the importance of inflammation in human diseases, there is an increasing appreciation in elucidating the role of lncRNAs in autoimmune diseases in order to use them as a diagnostic or prognostic marker. Beside many other diseases, lncRNA research is in primitive stage in aging-related disorders and syndromes. However, significant research has been dedicated to understanding the principles and functions of lncRNAs in senescence. Recently, senescence has been accepted as a beneficial and detrimental process to the well-being of an organism. In early developmental stages, senescence acts as a tumor suppressor mechanism and aids in tissue homeostasis. However, at older ages of an organism, senescence unleashes detrimental pro-inflammatory responses and diminishes immune function and participates in tumor initiation process. Hence, understanding of lncRNA-mediated or regulated cellular networks and signaling pathways in the right cellular context or precise developmental stages will aid us to develop specific pharmacological interventions against senescence-mediated anomalies such as cancer, cardiac, neurological and autoimmune human diseases.

Senescent cells display a flattened cell shape [104,105,106] and elevated senescence-associated β-galactosidase (SA-β-gal) activity, a gold standard for the detection of senescence [107,108]. However, several proliferating cells in skin and the duodenum exhibit non-specific SA-b-gal activity [109,110]. In contrast, cells from patients with autosomal recessive G(M1)-gangliosidosis lack lysosomal β-gal activity, but retain their ability to senesce [111]. These limitations warrant a finding of an alternative markers for senescence. p16^INK4a^- expression was demonstrated to be a promising marker due to its increased activity in senescence and aging [112,113,114,115,116,117,118]. However, the caveat stems from its similar expression pattern in preneoplastic lesions. This exception limited the scope of p16^INK4a^ usage as a *bona fide* senescence marker [114,119]. Loss of lamin B1 protein was suggested to be a promising biomarker for senescence but its mRNA levels failed to distinguish senescence vs quiescent fibroblasts [120]. Several groups came up with various senescence-associated markers such as γ-H2A-X and 53BP1 foci [121], activated ataxia-telangiectasia mutated (ATM) kinase [122], telomere-associated DNA damage foci (TAF) [116,123,124,125], and a histone variant H2A.J [126]. Most of them were either employed individually or in combination, but the identification of a unique senescent cell-specific marker remains unsolved [127]. Hence, future investigations are needed to characterize and identify senescence specific markers that are indispensable to reliably detect and quantify senescent cells in vitro and in vivo. So far, the majority of senescence markers are protein based. Several decades ago, a similar trend was observed in cancer biomarker research, but with the advent of high-throughput sequencing, non-coding RNAs evolved as a potential biomarker for various cancers. Hence, applying the similar strategy by identifying, annotating and exploring the novel lncRNA signatures across large set of senescence and aging models will allow us to decipher a quantitative and qualitative marker for senescence in vitro and in vivo.

Fibroblast undergoing replicative senescence demonstrated striking activation of retrotransposons and satellite repeats [128]. De Cecco et al. reported that replicative senescent cells exhibited global chromatin alteration resulting in gene silencing and activation of transposable elements (TEs). The authors proposed that such uncontrolled activation might lead to DNA damage response and genome instability. However, whether the activation of TEs in senescence is a consequence or contribution remains unanswered. Sequencing studies showed many of the transposable elements are located in the promoter regions of lncRNAs [129]. Functional studies suggest that activation of TEs promotes lncRNA expression in senescence and aging. Wang et al., [130] demonstrated that de novo lncRNAs from *Alu* TEs are over expressed during adult human stem cell aging. Loss of function of *Alu* lncRNAs reverse senescence, suggesting that TE derived lncRNAs directly promote senescence. Hence, detailed study of interrelationships between TEs and lncRNAs in different senescence models will allow us to identify the new functions of genomic dark matter in cancer and aging. Nuclear architecture undergoes dramatic alteration during aging. Nuclei of Hutchison-Gilford progeria syndrome, the Werner syndrome and ataxia telangiectasia patient cells share dramatic higher order chromatin alterations [131]. lncRNA is part of epigenetic mediated chromatin organization; it is tempting to speculate that the alterations of nuclear architecture in aging cells might be a direct consequence of lncRNA malfunctions. Insights from studies on MALAT1 and FIRR show that lncRNAs have the ability to reposition the large genomic regions and nuclear bodies [109,132]. It is plausible that distribution and stoichiometry of lncRNAs in senescence can have major effect in aging-associated nuclear disorganization and gene expression. Hence, rigorous functional characterization of lncRNAs by harnessing the CRISPR technology and single cell transcriptomics is needed to tease out the reactive, compensatory or causative role of lncRNAs nuclear organization in senescence.

Elucidation of the molecular mechanism through which lncRNAs function in senescence and aging disorders is likely to provide new understanding of lncRNAs in age-related diseases and uncover new opportunities for better treatment. With this summary, it is now evident that age-related diseases exhibited altered expression of lncRNAs that can be used as useful information for diagnosis. Hence, there is an immediate need to uncover novel functions of lncRNAs that modulate the expression of senescence regulators and the reexamination of senescence-specific lncRNAs may uncover disease-specific roles. Ongoing clinical trials of lncRNAs in cancer highlight the prospects of lncRNAs as new age molecules in senescence-associated human diseases.

## Figures and Tables

**Figure 1 ijms-20-02615-f001:**
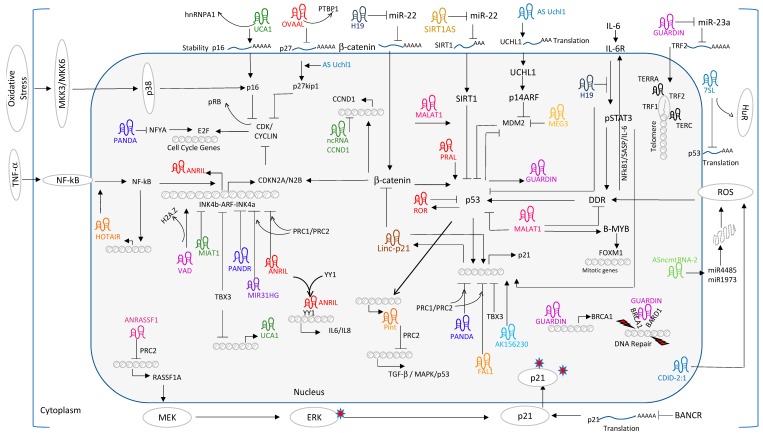
lncRNAs affect senescence pathways. External and internal factors trigger intracellular signaling pathways; these include MAPK, NF-kB, p53/p21, Rb/p16, IL-6/STAT3, b-catenin, and DDR. Activation of these pathways initiates cell cycle inhibition and promotes pro-senescence markers and inflammatory response. lncRNAs function at different stages either by activator or inhibitor of gene regulation.

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
