# Peer review of "LncRNAs Regulatory Networks in Cellular Senescence"

_ijms, 2019, doi:10.3390/ijms20112615_

Reviewer 1 Report

Remarks to the Author:

In this review, the author focus on lncRNAs that have been shown to play functional roles in cellular senescence and provide a comprehensive understanding of lncRNA-mediated molecular mechanisms in gene expression leading to senescence. In my opinion, a review on lncRNAs and senescence-associated pathways is still missing, as previous reviews have primarily focused on lncRNAs and age-associated diseases such as cardiovascular and neurodegenerative diseases. However, in order to be supportive of its publication, I recommend the author to revise the manuscript accordingly to the following major concerns.

 Major comments

1. The abstract is misleading, as it does not represent the major focus of this review. The review describes the lncRNAs so far implicated in senescence-associated pathways, but with no particular focus on cardiac diseases (actually the author most often describes cancer disease). Anyway, I consider this is an advantage, as a previous revision article on this topic has just been published (Bink DI et al., Noncoding RNA 2019). So, I recommend the author to revise the abstract.

2. The article should include figures summarizing the known roles of lncRNAs in each senescence pathway: pRb/p16, p53/p21, telomere erosion and stress response. The amount of information is too descriptive and needs figure support in order to keep the reader’s attention.

3. The article needs extensive English revision. I provide a pdf file including some suggestions.

4. The Conclusions and future directions section is too elusive. The author should provide clear input into the relevance of lncRNAs in senescence, examples on how this has been explored in therapy, biomarkers thus far well established, and what should be studied next, major questions unsolved.

5. ‘Other lncRNAs’ section should include an introductory paragraph explaining why do these lnRNAs are described separately from the other sections.

Author Response

Reviewer # 1:

1. The abstract is misleading, as it does not represent the major focus of this review. The review describes the lncRNAs so far implicated in senescence-associated pathways, but with no particular focus on cardiac diseases (actually the author most often describes cancer disease). Anyway, I consider this is an advantage, as a previous revision article on this topic has just been published (Bink DI et al., Noncoding RNA 2019). So, I recommend the author to revise the abstract. 

I agree and thank the reviewer for this suggestion. I revised the abstract and as per the reveiwers suggestion.     

2. The article should include figures summarizing the known roles of lncRNAs in each senescence pathway: pRb/p16, p53/p21, telomere erosion and stress response. The amount of information is too descriptive and needs figure support in order to keep the reader’s attention.

Thank you for pointing out. Our intention is to provide a global picture of how different lncRNAs regulate various signaling pathways in senescence. So, I integrated all the functional mechanism of lncRNAs and their downstream pathways and represented in a network illustration. I believe that such kind of snapshot will provide a detailed information about how these lncRNAs cross talk with each other through various direct and indirect feedback loops and function in unison to maintain senescence. 

3.  The article needs extensive English revision. I provide a pdf file including some suggestions.

Sincerely appreciate reviewer help and I incorporated all the suggestion mentioned in the PDF document. It helped me tremendously to clear all the mistakes 

4. The Conclusions and future directions section is too elusive. The author should provide clear input into the relevance of lncRNAs in senescence, examples on how this has been explored in therapy, biomarkers thus far well established, and what should be studied next, major questions unsolved.

I agree and provided several sections describing the potential of lncRNAs as biomarkers, what needs to be done and what are the caveats of existing studies of lncRNAs and what else can be done in senescence research.

5. â€˜Other lncRNAs’ section should include an introductory paragraph explaining why do these lnRNAs are described separately from the other sections.

As per the suggestion, an introductory paragraph is added at the begging of the "other lncRNAs" section

Reviewer 2 Report

In this manuscript, the author reviewed and summarized the current findings of the lncRNAs and their dysregulated signaling pathways in cardiac diseases. The topci is interesting. However, there are some problems in this manuscript. Hence, I would like to recommend major revision of this manuscript:

1. There were some grammatical errors in the article, and the expression of some of the content was not clear enough. The author needed to check the manuscript carefully and made corresponding revision.

2.The author should give some discussions of the recent trend of developing computational models to identify disease-lncRNA associations as future direction of this topic at least in the discussion section. Some studies are recommended and should be discussed (PMIDs:24002109, 26577439, 26278472, and 26061969)

3. Long non-coding RNA (lncRNA) plays an important role in many important biological processes and has attracted widespread attention.LncRNAs usually perform their functions by interacting with the corresponding RNA- binding proteins.The author should also discuss the recent trend of developing computational models to identify lncRNA-protein association in the discussion section. Some recommended studies are helpful (PMIDs: 30388620, 30023002, 29701758, 29583068 and 29262614).

4. The sentence of "LncRNAs are endogenously encoded single-stranded RNAs with more than 200 nucleotides that lack protein coding potential. lncRNAs exhibit tissue specificity, regulated expression pattern, unique mode of function and plays important roles in cell proliferation, cell fate decision, apoptosis, differentiation, stem cell maintenance and division. “ should have references of papers with PMIDs of 27345524 and 30247501.

5. Some figures should be provided to summarize the current findings of the lncRNAs and their dysregulated signaling pathways in cardiac diseases and enumerate the implication of lncRNAs as a therapeutic targets or biomarkers for different types of human cardiac disfunctions.

6. The novelty and limitations of the review relative to previous reviews about lncRNA should be provided.

Author Response

Reviewer 2:

In this manuscript, the author reviewed and summarized the current findings of the lncRNAs and their dysregulated signaling pathways in cardiac diseases. The topci is interesting. However, there are some problems in this manuscript. Hence, I would like to recommend major revision of this manuscript:

1. There were some grammatical errors in the article, and the expression of some of the content was not clear enough. The author needed to check the manuscript carefully and made corresponding revision.

I thank reviewer and incorporated several modifications throughout the article. 

2.The author should give some discussions of the recent trend of developing computational models to identify disease-lncRNA associations as future direction of this topic at least in the discussion section. Some studies are recommended and should be discussed (PMIDs:24002109, 26577439, 26278472, and 26061969)

I agree with the reviewer and added separate section on computational models and their usage and limitation in conclusions and future directions. All the above references are added. 

3. Long non-coding RNA (lncRNA) plays an important role in many important biological processes and has attracted widespread attention.LncRNAs usually perform their functions by interacting with the corresponding RNA- binding proteins.The author should also discuss the recent trend of developing computational models to identify lncRNA-protein association in the discussion section. Some recommended studies are helpful (PMIDs: 30388620, 30023002, 29701758, 29583068 and 29262614).

I discussed as per the reviewer suggestion and added all the above references in the manuscript. 

4. The sentence of "LncRNAs are endogenously encoded single-stranded RNAs with more than 200 nucleotides that lack protein coding potential. lncRNAs exhibit tissue specificity, regulated expression pattern, unique mode of function and plays important roles in cell proliferation, cell fate decision, apoptosis, differentiation, stem cell maintenance and division. “ should have references of papers with PMIDs of 27345524 and 30247501.

Added the above-mentioned references in manuscript. 

5. Some figures should be provided to summarize the current findings of the lncRNAs and their dysregulated signaling pathways in cardiac diseases and enumerate the implication of lncRNAs as a therapeutic targets or biomarkers for different types of human cardiac disfunctions.

I sincerely apologize for the confusion. I would like to clarify that this review described about the functions of lncRNAs in senescence regulation but not cardiac diseases. So as reviewer suggested, I prepared a fig illustrating the functions of lncRNAs and their downstream signaling pathways in senescence. 

6. The novelty and limitations of the review relative to previous reviews about lncRNA should be provided.

Briefly described in the abstract. 

Round  2

Reviewer 2 Report

The authors have addressed my comments and I agree acceptance.